# Endothelial Progenitor Cells as Pathogenetic and Diagnostic Factors, and Potential Targets for GLP-1 in Combination with Metabolic Syndrome and Chronic Obstructive Pulmonary Disease

**DOI:** 10.3390/ijms20051105

**Published:** 2019-03-04

**Authors:** Evgenii Germanovich Skurikhin, Olga Victorovna Pershina, Angelina Vladimirovna Pakhomova, Edgar Sergeevich Pan, Vyacheslav Andreevich Krupin, Natalia Nicolaevna Ermakova, Olga Evgenevna Vaizova, Anna Sergeevna Pozdeeva, Mariia Andreevna Zhukova, Viktoriia Evgenevna Skurikhina, Wolf-Dieter Grimm, Alexander Mikhaylovich Dygai

**Affiliations:** 1Laboratory of Regenerative Pharmacology, Goldberg ED Research Institute of Pharmacology and Regenerative Medicine, Tomsk National Research Medical Centre of the Russian Academy of Sciences, Tomsk 634028, Russia; eskurihin@inbox.ru (E.G.S.); ovpershina@gmail.com (O.V.P.); artifexpan@gmail.com (E.S.P.); vakrupin88@gmail.com (V.A.K.); nejela@mail.ru (N.N.E.); amdygay@gmail.com (A.M.D.); 2Siberian State Medical University, Tomsk 634050, Russia; vaizova@mail.ru (O.E.V.); Katapylca@yandex.ru (A.S.P.); mashazyk@gmail.com (M.A.Z.); vskurikhina@mail.ru (V.E.S.); 3Periodontology, Department of Dental Medicine, Faculty of Health, University of Witten/Herdecke, 58455 Witten, Germany; prof_wolf.grimm@yahoo.de

**Keywords:** chronic obstructive pulmonary disease, obesity, metabolic syndrome, endothelial progenitor cells, glucagon-like peptide 1

## Abstract

In clinical practice, there are patients with a combination of metabolic syndrome (MS) and chronic obstructive pulmonary disease (COPD). The pathological mechanisms linking MS and COPD are largely unknown. It remains unclear whether the effect of MS (possible obesity) has a major impact on the progression of COPD. This complicates the development of effective approaches for the treatment of patients with a diagnosis of MS and COPD. Experiments were performed on female C57BL/6 mice. Introduction of monosodium glutamate and extract of cigarette smoke was modeled to simulate the combined pathology of lipid disorders and emphysema. Biological effects of glucagon-like peptide 1 (GLP-1) and GLP-1 on endothelial progenitor cells (EPC) in vitro and in vivo were evaluated. Histological, immunohistochemical methods, biochemical methods, cytometric analysis of markers identifying EPC were used in the study. The CD31^+^ endothelial cells in vitro evaluation was produced by Flow Cytometry and Image Processing of each well with a Cytation™ 3. GLP-1 reduces the area of emphysema and increases the number of CD31^+^ endothelial cells in the lungs of mice in conditions of dyslipidemia and damage to alveolar tissue of cigarette smoke extract. The regenerative effects of GLP-1 are caused by a decrease in inflammation, a positive effect on lipid metabolism and glucose metabolism. EPC are proposed as pathogenetic and diagnostic markers of endothelial disorders in combination of MS with COPD. Based on GLP-1, it is proposed to create a drug to stimulate the regeneration of endothelium damaged in MS and COPD.

## 1. Introduction

Metabolic syndrome (MS) is a complex clinical condition, and obesity is considered to be one of its main components [1]. The main criterion for the diagnosis of metabolic syndrome is the presence of abdominal-visceral obesity in the patient [2]. Approximately 20–30% of the adult population worldwide is obese [3], and its prevalence is constantly growing [4]. Obesity significantly increases the risk of cardiovascular disease, it is the main cause of insulin resistance and the development of type 2 diabetes, and it increases skeletal muscle dysfunction and obstructive sleep apnea [5,6,7]. Obesity is noted to promote an increase in respiratory reactivity, and this can lead to the development of various respiratory pathologies, including chronic obstructive pulmonary disease (COPD), asthma and other lung diseases [8,9]. It is believed that 10% of obese patients suffer from COPD [10,11]. A complication of obesity such as type 2 diabetes can worsen the course of COPD and promote the progression of COPD, increasing the likelihood of death of patients [12,13,14]. Moreover, obesity is common in patients with COPD and contributes to respiratory symptoms [15].

Systemic inflammation is referred to as a common feature of the metabolic syndrome and COPD [16]. Thus, the adipose tissue of patients diagnosed with obesity contributes to inflammation by releasing proinflammatory mediators in the bloodstream [17]. On the other hand, in COPD there is an increase in the concentration of proinflammatory mediators (IL-1, IL-6, TNF-α) and adipose tissue hormones (leptin, adiponectin and resistin) in the bloodstream, and this can lead to insulin resistance, and chronic hyperglycemia. Furthermore, glycation products increase synthesis and deposition of collagen in the lungs [18].

Vascular complications often develop in patients with MS and obesity, and there is a decrease in the number of circulating endothelial cells and disorders of their functions [19]. On the other hand, endothelial dysfunction plays a major role in the development of vascular diseases associated with pulmonary circulation [20]. There is a violation of the endothelial function of pulmonary arteries in patients with COPD [21]. The production of reactive oxygen species can act as a general mechanism of endothelial damage, dysfunction, and endothelial cell apoptosis in obesity and COPD [22,23].

Nowadays, the degree of therapy development in treatment of disorders of endothelium of pulmonary vessels in chronic obstructive pulmonary disease burdened by obesity is insignificant. The lack of a solution to this problem is due to the difficulty of choosing a potentially pharmacologically active molecule and a potential target. We turned our attention to the hormone glucagon-like peptide-1 (GLP-1). GLP-1 (GLP-1R) incretin receptor is a membrane receptor associated with G-protein, found not only in the cells of the islets of Langerhans in the pancreas, but also in the central nervous system, kidneys, heart, blood vessels, lungs; it is expressed on endothelial and neuronal cells [24,25,26]. In addition to endocrine effects, GLP-1 has an anti-inflammatory effect [26,27], and it improves microvascular perfusion [28]. The mechanisms by which GLP-1 can have anti-inflammatory and microvascular effects are unclear. GLP-1 is thought to prevent dysfunction and autophagy in endothelial cells [23]. The effect of the hormone on endothelial progenitor cells (EPC) is not excluded. Participation of EPC in regeneration of endothelium of the lungs in simulation of emphysema [29], pancreas and testicles in obesity simulation [30] was shown by us in earlier works.

The aim of this study was to examine the effectiveness of the GLP-1 treatment in inflammation and emphysema, incretin stimulation of endothelium and epithelium regeneration of the alveolar tissue in the C57BL/6 mice with lipid disorders and emphysema. We investigated the effect of GLP-1 on bone marrow, circulating in the blood and tissue endothelial progenitor cells in order to discuss possible mechanisms of the hormone’s action.

## 2. Results

### 2.1. Changes in Serum Biochemical Parameters in MSG-Treated Mice Prior to Pulmonary Emphysema Modeling

To confirm obesity and hyperglycemia in the female C57BL/6 mice, mice that received MSG (monosodium glutamate) were analyzed for triglycerides (TG), high-density lipoproteins (HDL) and low-density lipoproteins (LDL) in blood serum and the concentration of glucose in the blood on the 124th day of the experiment. As a result, of MSG administration, there was an increase in the concentration of TG (by 229%), LDL (by 142%) and atherogenicity index by 182% in mice under the management of sodium glutamate in relation to intact control. In contrast, the concentration of HDL decreased (by 36.5%) (Figure 1). In mice under the management of sodium glutamate hyperglycemia was noted. Thus, before modeling emphysema in mice, their fat metabolism was impaired.

### 2.2. Effect of GLP-1 on Serum Lipid Profile

On the 188th day of the experiment, we studied the effectiveness of GLP-1 on fat metabolism in mice in conditions of MSG and cigarette smoke extract (CSE) administration.

Before we considered the mice treated with sodium glutamate (group 2). On day 188, in group 2 mice there was an increase in serum concentrations of TG and LDL, and a decrease in HDL concentrations (Figure 1A–C). Similar changes were detected on the 124th day of the experiment. In group 3 mice (CSE) there were no significant changes in fat metabolism. In group 4 mice (obesity and CSE) the concentration of high-density serum lipids decreased (Figure 1C) in relation to the intact control.

Introduction of GLP-1 in MG and CSE treated animals (group 5) reduced LDL concentration (by 38%) and increased HDL concentration (by 53%) in relation to group 4 animals (Figure 1B,C).

### 2.3. Effect of GLP-1 on Blood Glucose, GTT and AUC

In mice of group 3, glucose concentration did not differ from that in mice of group 1. In group 2 and 4 mice, hyperglycemia was detected on day 124, 126, 147, and 188 of the experiment (Figure 2A). GLP-1 had no effect on hyperglycemia in group 5 mice.

GTT was conducted on the 186th day of the experiment. In mice of groups 2, 3 and 4 there was a high level of glucose from the 15th minute till the 90th minute of observation (Figure 2B). At the same time, glucose levels in mice of group 3 were significantly lower at the 90th minute of observation compared to those in groups 2 and 4. In group 5 mice, GLP-1 reduced glucose levels from the 60th until the 90th minute compared to group 4 (Figure 2B).

Additionally, on the 186th day of the experiment, the area under the curve (AUC) for blood glucose was calculated. According to our data, mice of group 2 (obesity) showed an increase in AUC by 45% (*p* < 0.05) compared to intact control; mice of group 3 (CSE) by 39.8% (*p* < 0.05); and mice of group 4 (obesity and CSE) by 48% (*p* < 0.05) (Figure 2C). Assignment of GLP-1 to group 5 mice significantly reduced AUC (by 16.5%) in relation to group 4 mice.

### 2.4. Effect of GLP-1 on Pathological Changes of the Lungs

Staining with hematoxylin and eosin revealed pathological changes in the lung tissue of mice on day 188 of the experiment. Inflammatory infiltration, edema, microcirculation disorders were not revealed in the lungs of intact control mice and mice with MSG (Figure 3A,B). In mice with CSE, venous stagnation, moderate inflammatory infiltration of the lung parenchyma (mainly peri-bronchial infiltrates) were observed. The alveolar walls thickened due to inflammatory infiltration. The composition of the infiltrate is represented by macrophages and lymphocytes (Figure 3C,D). With the combined effect (obesity and CSE), an increase in venous stagnation and inflammatory infiltration was noted (besides macrophages and lymphocytes in the inflammatory infiltrate, neutrophils were observed) compared to group 3, the walls of blood vessels and bronchi were thickened. Pathomorphological changes in the lungs of group 5 mice treated with GLP-1 were less pronounced than in group 4 (Figure 3E).

Study of the micro preparations of the lungs did not reveal emphysema of the lung in mice of group 2. In the upper, middle and lower regions of the lungs of groups 3 and 4 mice emphysema was observed. The introduction of GLP-1 significantly reduced the area of emphysematous dilated tissue in the lower region of the lungs of group 5 mice compared to group 4 (Figure 3F).

### 2.5. Immunohistochemical Examination of the Lungs

The introduction of CSE significantly reduced the number of cells expressing CD31, α1-antitrypsin and pan-cytokeratin (AE1/AE3) in the pulmonary tissue of mice (group 3) compared to intact control (group 1) on day 188 of the experiment (Figure 4). A similar decrease in the number of CD31^+^ cells, α1-antitrypsin^+^ cells, and AE1/AE3^+^ cells was observed in the lungs of mice under the administration of MSG and CSE (group 4). The study of insulin expression in the lung tissue of mice revealed no differences in groups 1, 3 and 4.

Treatment with GLP-1 caused a significant increase in the number of CD31^+^ cells and Insulin+ cells in the lungs of group 5 mice compared to group 4.

### 2.6. Flow Cytometric Analysis

#### 2.6.1. Endothelial Progenitor Cells

On day 188 of the experiment, endothelial progenitor cells with phenotype CD31^+^CD34^+^CD146^+^ and CD45^−^CD31^+^CD34^+^, precursors of angiogenesis (CD45^−^CD309^+^CD117^+^) were studied in the lungs, bone marrow and blood of mice (groups 1, 4 and 5). According to our data, EPC response to the modeling of combined pathology (obesity and emphysema) was different. Thus, the number of CD31^+^CD34^+^CD146^+^ EPC significantly decreased in bone marrow, blood of group 4 mice compared to intact control, the number of pulmonary EPC increased (CD31^+^CD34^+^CD146^+^) (Figure 5). The evaluation of CD45^−^CD31^+^CD34^+^ EPC revealed a decrease in their number in the blood of group 4; on the contrary, in the bone marrow and lungs, their number increased. When modeling a combined pathology, we did not observe changes in bone marrow progenitors of angiogenesis, but their numbers in the lungs significantly exceeded those in group 1.

GLP-1 reduced the number of angiogenesis precursors in the lungs (by 53%) and bone marrow (by 41%) of group 5 mice relative to group 4 mice (Figure 5). On the other hand, the therapy did not affect the content of CD31^+^CD34^+^CD146^+^ EPC in the blood and lungs of animals with comorbidities, but increased their number in the bone marrow. In group 5, we observed a significant (1.8 times) expansion of circulating EPC in the blood by the phenotype CD45^−^CD31^+^CD34^+^ in comparison with group 4, in the lungs the number of these cells decreased by 72% (Figure 5).

#### 2.6.2. Inflammatory Cells

Modeling of combined pathology did not affect CD3^+^ T lymphocytes and increased the number of F4/80^+^ macrophages (by 52%) and CD11b^+^ macrophages (by 43%) in the blood of mice compared to intact control (Figure 6). Treatment of GLP-1 had a positive effect on the content of inflammation cells in the blood of group 5 mice: there was a significant decrease in the number of CD3^+^ T lymphocytes (by 33%), F4/80^+^ macrophages (by 24%) and CD11b^+^ macrophages (by 26%) compared to group 4.

### 2.7. Effect of GLP-1 on Apoptosis of CD31^+^ Lung Cells in vivo

Programmable death (apoptosis) was studied in CD31^+^ endothelial cells obtained from lungs of C57BL/6 group 1, group 4 and group 5 mice on day 188 of the experiment. Modeling of combined pathology (obesity and emphysema) significantly increased (by more than 3 times) the number of CD31^+^ endothelial cells in apoptosis compared to the control group 1.

Course administration of GLP-1 reduced the number of apoptotic CD31^+^ endothelial cells in the lungs of mice in conditions of combined pathology by 30% compared to untreated mice of group 4 (Figure 7A).

### 2.8. Effect of GLP-1 on CD31^+^ Lung Cells in vitro

In vitro, the effect of GLP-1 on some parameters of CD31^+^ cells obtained from the lungs of group 1 and group 4 mice was studied. GLP-1 was introduced into the enriched population of CD31^+^ endothelial cells, the final concentration of incretin in the culture was 10^−7^ M. In Figure 7D,E it can be seen that under the action of incretin, the number of apoptotic CD31^+^ cells in group 4 decreased significantly (more than 2 times compared to the culture without incretin), the number of CD31^+^ cells in group 1 did not change.

GLP-1 significantly increased CD34 marker expression in CD31^+^ culture of endothelial lung cells in group 1 and group 4 (Figure 7B).

GLP-1 increased the number of CD31^+^ endothelial cells with active esterases. This effect of incretin was more pronounced in group 1 than in group 4 (Figure 7C).

## 3. Discussion

As the problem of obesity grows in modern society, its influence on the state of many organs, including the lungs, is becoming more and more obvious [8]. The results of clinical studies of the interaction between obesity and concomitant MS with lung diseases such as asthma, pneumonia, acute respiratory distress syndrome (ARDS) and COPD are often contradictory [8,9]. Therefore, there is a need for experimental approaches to understand common pathological mechanisms of obesity and lung disease.

There is evidence that the introduction of MSG to newborn mice damages the arcuate nucleus of the hypothalamus and reduces the level of neuropeptide Y, which regulates the transmission of signals of leptin and insulin [31]. These animals develop severe obesity and diabetes mellitus (with hyperglycemia and hyperinsulinemia) [32]. There is a similarity of MSG effects in rodents with the clinical course of MS in humans [32,33]. In our experiments, MSG-treated female C57BL/6 mice showed an increase in Lee index, concentration of serum triglycerides and low-density lipoproteins, hyperglycemia, and impaired glucose tolerance on the 124th day of the experiment (Figure 1). As you can see, MSG induces obesity in mice and forms diabetic symptoms, which can be interpreted as MS.

COPD is a consequence of chronic inflammation and trauma of the alveolar endothelium, arising in response to the action of nicotine and resins. We reproduced such a variety of etiological factors of COPD introducing lipopolysaccharide (LPS) and cigarette smoke extract (CSE). On the 147th day of the experiment, inflammation and diffuse emphysema of the lungs, damage to the alveolar epithelium and endothelium, and a decrease in the concentration of alpha1-antitrypsin were revealed in mice with metabolic disorders, treated with LPS and CSE. Similar pathological changes in mice treated with CSE have been published previously [34]. In the present study, group 4 mice retained dyslipidemia, hyperglycemia and violation of GTT, and pathological deposition of visceral fat (the results are not presented).

In one of the previous studies, a decrease in the number of circulating endothelial cells (CD34^+^CD309^+^ and CD34^+^CD309^+^CD133^+^) was found in patients with COPD [35]. According to Doyle M. F. with co-authors (2017), these data point to the violation of endothelium regeneration of the lungs in patients with COPD. However, the EPS population is not homogeneous. EPC is defined as CD34^+^VEGFR2^+^CD45^dim^ [36]. Other authors use coexpression of CD34 and fibroblast growth factor receptor 1 to identify EPC [37]. As you can see, positions on the phenotype of EPC for pulmonary diseases are still unsettled. Complicating the situation is the fact that COPD is often the result of a variety of etiological factors. From these positions, it is quite natural that the nature of COPD could determine the mechanisms of alveolar tissue regeneration. We received confirmation of this in our earlier work. According to our data, emphysema of different etiological factors (elastase, cigarette smoke extract, D-galactosamine hydrochloride, tyrosine kinase inhibitor SU5416) mobilizes EPC with the phenotype CD45^−^CD309^+^, CD45^−^CD31^+^CD34^+^ and/or CD45^−^CD309^+^CD117^+^ in regeneration of the alveolar endothelium [34].

The results of flow cytometric analysis of this work indicate that phenotypically different EPC (CD45^−^CD309^+^CD117^+^; CD45^−^CD31^+^CD34^+^; CD31^+^CD34^+^CD146^−^) were recruited from the bone marrow to the injured lungs of mice with a combined pathology (MS and pulmonary emphysema) on the 146th experiment day (results not shown) with preservation of the pattern for 188 days. In his report, Agusti et al. (2013) indicates the direct dependence of the recruitment of EPC on inflammatory cytokines [21]. It is possible that the selective recruitment of EPC into the mice lungs in conditions of MSG and CSE introduction is directly dependent on the secretory activity of inflammatory cells. However, despite an increase of EPC in the lung tissue endothelial damage of the mice lungs in conditions of combined pathology persists. Perhaps this is due to the violation of intercellular contacts, transendothelial transition and differentiation of EPC. As is known, MSCs are involved in restoring the normal structure and function of damaged lungs. The decrease in the number of MSCs in the lungs of mice in conditions of combined pathology partially explains the lack of consistency of the mechanisms of endothelial regeneration (results not shown).

Thus, in conditions of MS and COPD, endothelial progenitor cells can act as specific markers of pathological changes in microcirculation and destruction of the alveolar endothelium. On the other hand, they can act as markers of regeneration of lung endothelium. However, while conducting a quantitative assessment of EPC, it is necessary to consider the nature and factors complicating the disease, and to conduct compulsory additional studies in vitro to evaluate the potential for EPC regeneration.

Complicated combined pathologies are difficult to treat, so the search for drugs that can influence the pathogenetic basis of the disease is very important. GLP-1 is known to stimulate insulin production by islet β-cells, it counteracts insulin resistance, improves tolerance to peripheral glucose, has anti-inflammatory properties [26,27]. In addition to endocrine activity, GLP-1 can play a crucial role in lung homeostasis. GLP-1 receptors are abundantly present in the alveoli, septum, airway and vascular smooth muscle [24,25,38], their levels are relatively higher than in the intestine and brain [39]. There is no evidence of GLP-1 production in the lungs [40]. From this point of view, we attempted to treat the combined pathology modeled by us with neuropeptide and incretin GLP-1 from the 147th till the 187th day of the experiment.

GLP-1 introduction naturally prevents hyperglycemia, while in the vessels of group 5 mice lung tissue, a significant increase in the expression of insulin was found (the 188th day). Additionally, the treated animals showed an improvement in serum lipid metabolism and a decrease in visceral fat (Figure 1). On the other hand, GLP-1 shows anti-inflammatory activity, as evidenced by the results of a study of inflammatory infiltration on lung preparations stained with hematoxylin and eosin, and a decrease in the number of inflammatory cells circulating in the blood (CD3^+^ T-lymphocytes, F4/80^+^ macrophages, CD11b^+^ macrophages) (Figure 3 and Figure 6).

All these results of treatment of mice with MS and pulmonary emphysema were expected for us. They are explained by the known ability of GLP-1 to enhance insulin secretion by β-cells, to restore insulin sensitivity of cells and glucose uptake by tissues, and to normalize fat metabolism. More unexpected for us was the reduction in the area of emphysematous-expanded alveolar tissue of mice with a combined pathology affected by GLP-1 (Figure 3). When discussing this issue, we turned our attention to the well-known high protease activity of inflammatory cells in the lung tissue in COPD [41,42]. As was shown above, incretin reduces the number of inflammatory cells, which naturally should lead to a decrease in the level of proteases in the lung tissue. On the other hand, a glycoprotein such as alpha1-antitrypsin is synthesized in the liver and inhibits the action of trypsin, chymotrypsin, elastase, kallikrein, cathepsins and other tissue protease enzymes. Alpha1-antitrypsin is deposited by alveolar cells. There are reports that indicate the positive effect of GLP-1 on hepatocytes [43]. Based on this, a slight increase in the expression of alpha1-antitrypsin in the lungs of treated mice with a combined pathology may be associated with hepatoprotective activity of incretin. Thus, a decrease in the area of emphysematous-expanded alveolar tissue in GLP-1-treated mice with combined pathology is the result of the inhibition of the protease mechanism of pulmonary emphysema formation.

In conclusion, it is necessary to dwell on the phenomenon of reducing the recruitment of CD45^−^CD309^+^CD117^+^ EPC and CD45^−^CD31^+^CD34^+^ EPC from the bone marrow to the damaged lung tissue against the background of increased expression of the endothelial marker CD31 in the lungs of group 5 mice compared to mice of group 4 (Figure 5). We associate a decrease in the activity of recruiting EPC with the anti-inflammatory activity of GLP-1. In turn, it is known that GLP-1 can prevent the effects of oxidative stress and autophagy [23]. This message prompted us to further evaluate apoptosis of CD31^+^ lung cells. As a result of these studies, we obtained evidence of the anti-apoptotic activity of incretin in its action on CD31^+^ lung endothelial cells of mice with MS and pulmonary emphysema in vivo and in vitro (Figure 7A,D). Attention is drawn to the fact that in the culture of CD31^+^ lung endothelial cells obtained from mice with MS and pulmonary emphysema (group 4), the number of cells expressing the intercellular adhesion molecule CD34 (Figure 7B) sharply increases under the influence of the direct action of GLP-1. The number of CD31^+^ cells with active esterases increases, too (Figure 7C). These in vitro results indicate that endothelial progenitor cells expressing markers of CD31 and CD34 endothelial progenitor cells may be targets for GLP-1 and are involved with incretin in the regeneration processes of damaged endothelium in mice with MS and pulmonary emphysema.

Utilizing our novel mouse model for microenvironment-specific responses of lung tissues, we tried to demonstrate that the relationship between visceral fat and lung diseases, and therefore in longevity, is obviously causal. Supporting our experimental data [44] several epidemiologic studies have implicated visceral fat as a major risk factor for insulin resistance, type 2 diabetes mellitus, cardiovascular disease, stroke, metabolic syndrome and death, while the accrual of subcutaneous fat does not appear to play an important role in the etiology of disease risk. Glucagon-like peptide-1 (GLP-1) is an important insulin secretagogue that possesses anti-inflammatory effects [45]. In this study, we presented evidence of the effectiveness of treatment with incretin GLP-1 for the symptoms of obesity and diabetes, inflammation and emphysema, as well as stimulation of regeneration of damaged endothelium of the lungs in conditions of modeling obesity and emphysema. We have proposed a potential target of GLP-1: expressing markers of CD31 and CD34 endothelial progenitor cells. GLP-1 receptor (GLP-1R) agonists have been demonstrated to serve a pivotal role in the treatment of obstructive lung diseases, including chronic obstructive pulmonary disease (COPD). However, the specific function and underlying mechanisms of GLP-1R in COPD and in MS (possible obesity) remain uncertain. The aim of the present study was to investigate the action and underlying mechanisms in our novel female C57BL/6 mice model for these diseases. The results of the present study suggested that GLP-1R contributes to microenvironment-specific responses of lung tissues, potentially via an ABCA1-mediated pathway [45]. These results also suggest that GLP-1R may be a potential therapeutic target for the treatment of COPD and of MS (possible obesity).

## 4. Materials and Methods

### 4.1. Animals

Experiments were carried out on female C57BL/6 mice (certified animals from the nursery of E. D. Goldberg Research Institute of Pharmacology and Regenerative Medicine) in strict adherence to the principles of European Convention for the Protection of Vertebrate Animals used for Experimental and other Scientific Purposes (Strasbourg, 1986). The study was approved by the Ethic Committee of E. D. Goldberg Research Institute of Pharmacology and Regenerative Medicine (protocol IACUC No. 114062016). The day of birth was considered as experimental day 0.

### 4.2. Obesity Experimental Design

Female offspring received a subcutaneous (sc) daily injection of monosodium glutamate (MSG; Sigma, St. Louis, MO, USA) diluted in buffer solution at a dosage of 2.2 mg/g body weight from the first to the tenth day of life [46]. Physiological saline was injected into control mice in equivalent volume. The obesity parameter was estimated according to the Lee index for each animal on the 124th day of the experiment [46,47,48]. This index, calculated as a cubic root of body weight (g) * 10/nasoanal length (mm), equal to or lower than 0.300 was classified as normal. Female mice with Lee index values higher than 0.300 were classified as obese and included in this experiment [49].

### 4.3. Exposure to Cigarette Smoke Extract

On the 126th day of the experiment lung emphysema was induced. Lung emphysema was caused by course intratracheal administration of lipopolysaccharide (LPS, Sigma, St. Louis, MO, USA) and cigarette smoke extract (CSE) [50,51,52]. LPS at a dose of 3 µg/mouse in 50 µL phosphate buffer and 50 µL CSE was administered intratracheally. With the introduction of LPS and CSE general anesthesia (pentobarbital) was used. LPS was administered on the 126th and 129th days of the experiment. CSE was introduced on the 127th, 130th, 133d, 136th, 139th, 142nd, 149th, 156th, 163d and 170th day of the experiment.

Previously, smoke extract (CSE) from cigarettes of brand L&M REDLABEL 2 cigarettes/mL was obtained (the composition of a cigarette: resin 10 mg/CIG, nicotine 0.8 mg/CIG, CO 10 mg/CIG). Before obtaining the extract, the cigarette filter was removed, the length of a cigarette with the filter was 80 mm, 55 mm with the removed filter. The extraction was carried out by stretching the smoke of a lit cigarette through the phosphate buffer at a constant speed with the help of a vacuum pump, the cigarette was burned to a length of 5 mm. The burning time of one cigarette was 180 s. To remove the particles, the extract was filtered through a bacterial filter with a pore size of 45 nm. To standardize the obtained extract, pH (pH~7) and optical density were measured at wavelengths of 405 and 540 nm (D405~237, D540~123) before and after filtration.

### 4.4. Test Preparation

Glucagon-like peptide-1 (GLP-1) is a neuropeptide and an incretin derived from the transcription product of the proglucagon gene (Sigma, St. Louis, MO, USA). GLP-1 binds with high affinity to G protein-coupled receptors (GPCRs) located on pancreatic beta cells, and it exerts insulinotropic actions that include the stimulation of insulin gene transcription, insulin biosynthesis, and insulin secretion. GLP-1 was daily administered intraperitoneally in the region of the pancreas in dose of 3 mmol/kg on the 147th–187th days of the experiment.

### 4.5. Distribution of Animals Into Groups

The non-obese mice treated with saline were divided into two groups: control mice (group 1—“control”, *n* = 9) and mice exposed to cigarette smoke extract (group 3—“CSE”, *n* = 10). Animals with a Lie index greater than 0.300 were divided into three experimental groups: mice with obesity (group 2—“obesity”, *n* = 10); mice with obesity and exposure to cigarette smoke extract (group 4—“obesity + CSE”, *n* = 10); and mice with obesity and exposure to cigarette smoke extract treated with GLP-1 (group 5—“obesity + CSE+GLP-1”, *n* = 10). All mice were removed from the experiment on day 188 after birth by CO_2_ overdose.

### 4.6. Blood Glucose Determination and Glucose Tolerance Test (GTT)

Blood glucose level was determined using a glucometer (Accu-Chek Performa Nano (Roche Diagnostics GmbH, Mannheim, Germany) on the 124th, 126th, 147th and 188th days of the experiment. On the 186th day of the experiment, a glucose tolerance test was performed. Measurement of the initial level of glucose in the blood of animals was performed after 16 h of feed deprivation. After that, an intragastric administration of glucose (D-glucose, Sigma, St. Louis, MO, USA) was administered at a dose of 2 g/kg. Blood samples for studies of glucose were taken in 15, 30, 60 and 90 min after glucose administration [53]. Glucose responses during the glucose tolerance test were estimated from the total area under the curve (AUC).

### 4.7. Lipid Profile Determination

The lipid profile was determined on the 124th and 188th days of the experiment. Blood samples were taken from each animal into tubes without additives, kept at room temperature for 30 min and then centrifuged at 300 g for 10 min. Serum was separated and used to study lipid profile parameters. The concentration of cholesterol and triglycerides was determined by direct enzymatic methods using BioSystems reagents (Barcelona, Spain) in accordance with the instructions for use. Fractions of high-density lipoproteins (HDL) and low-density lipoproteins (LDL) were precipitated with phospholphramate and polyvinyl sulfate respectively, and then their concentration was determined by the level of residual cholesterol. All the results were expressed in mmol/L.

### 4.8. Morphological Examination of Lungs

The morphological examination of lungs was performed on day 188 of the experiment. To do this, the left lobe of the lung was fixed in a 10% solution of neutral formalin, carried out through alcohols of ascending concentrations to xylene and poured into paraffin according to the standard procedure. 5 µm thick dewaxed cuts were stained with hematoxylin and eosin [29]. Micro-preparations from each experimental animal were examined under the light microscope Axio Lab.A1 (Carl Zeiss, MicroImaging GmbH; Göttingen, Germany) at 100 × and 400 × magnifications. Microenvironment-specific responses of lung tissue, the presence of edema and inflammatory infiltration, venous congestion, thickening of vessel walls and bronchi were assessed [54,55,56].

### 4.9. Immunohistochemical Examination of the Lungs

Immunohistochemical examination of the lungs was performed on day 188 of the experiment. Sections of lung tissue were placed on glass slides with an adhesive polylysin coating (Leica Biosystems, Nussloch, Germany). Before staining, tissue sections were dewaxed with the following antigen unmasking in a citrate buffer (pH = 6) for 20 min. Incubation with primary antibodies was performed in a wet chamber at 37 °C. The following primary markers were used to identify specific cell markers antibodies: polyclonal antibodies to insulin (ab63820, Abcam, Cambridge, MA, USA), polyclonal antibodies to membrane protein CD31 (ab28364, Abcam, USA), polyclonal antibodies to α1-antitrypsin (ab9373, Abcam, Cambridge, MA, USA), monoclonal antibodies to pan-cytokeratin (AE1/AE3) (ab80826, Abcam, Cambridge, MA, USA). For antibody detection, an imaging system was used in accordance with the manufacturer’s instructions (Spring bioscience, Pleasanton, CA, USA). The contrast of the sections was performed with hematoxylin. After staining, the slices were dehydrated in xylene and enclosed in the installer environment. The microscope Axio Lab.A1 (Carl Zeiss, MicroImaging GmbH; Göttingen, Germany) with the AxioCam ERc5s camera (Carl Zeiss, Göttingen, Germany) was used to obtain micrographs. The analysis of the obtained images and the counting of cells expressing detectable antigens was performed using the ImageJ program (Madison, WI, USA).

### 4.10. Flow Cytometric Analysis

Mononuclear cells from blood, bone marrow, and lung were obtained by classical methods as was described earlier [34,57]. Membranes receptor expression of murine mononuclear cells derived from bone marrow, blood, and lung were analyzed using BD surface markers (BD Biosciences, San Jose, CA, USA). Single cell suspensions were stained with fluorophore-conjugated monoclonal antibodies: CD45 PerCP, CD31 APC, CD34 FITC, CD146 PerCP-Cy5.5, CD3 APC-Cy7, F4/80 BV424, CD11b AF647, CD309 (Flk-1) APC, CD117 PeCy7. The relevant isotype controls were used. Labeled cells were thoroughly washed with PBS and analyzed on FACSCanto II (Becton Dickinson, San Jose, CA, USA) with the FACS Diva software. A minimum of 100,000 events were recorded for each condition.

### 4.11. Lung Tissue Dissociation, Isolation and Magnetic Separation of CD31^+^ Lung Endothelial Cells

On the 188th day of the experiment, we studied the effect of GLP-1 on CD31^+^ lung endothelial cells of mice of the control group (group 1) and mice with obesity and pulmonary emphysema (group 4) in vitro. To do this, we isolated the lungs of mice of groups 1 and 4; lung tissue was cut into pieces, cleaved with a solution of collagenase/dispase (StemCell Technologies, Vancouver, BC, Canada) and mechanically dispersed into a suspension of individual cells. After removing the supernatant, the cell pellet was washed once with complete DMEM medium, and then resuspended in 10 mL complete DMEM medium and placed in a gelatin-coated plastic tissue culture plate T-25 for 24 h. The next day, the cells were removed from the plate using trypsin for magnetic sorting.

Magnetic sorting to enrich the cell suspension of lung CD31^+^ with endothelial cells was performed as standard using positive selection with anti-CD31 antibodies (StemCell Technologies, Canada). The fraction of CD31^+^ cells was isolated using EasySep^TM^. The saturated suspension of CD31^+^ cells (10^6^ cells/1 mL of medium) obtained after magnetic sorting was cultured on gelatin-coated plastic tablets for T-25 cell cultures in the medium M199 for 5 days in standard gas (3.5% CO_2_) and temperature conditions (37 °C). The medium was changed every 1–2 days. Evaluation of the efficiency of cell separation EasySep ™ and the effectiveness of the cultivation carried out to increase the mass of CD31^+^ cells was performed using flow cytometry.

### 4.12. Cultivation of CD31^+^ Lung Endothelial Cells with GLP-1

After a 6-day CD31^+^ culture cycle, the cells of group 1 and 4 mice were removed from the plate using trypsin, the M199 medium was changed, and the cells were sown at a concentration of 3 × 10^5^ cells/1 mL of medium on gelatin-coated plastic T-25 tablets. Before cultivation on the medium with CD31^+^ cells, we added GLP-1 (10^−7^ M). The cultivation cycle under standard conditions (3.5% CO_2_, 37 °C) was 24 h. At the end of the cultivation, the effectiveness of GLP-1 on CD31^+^ cells was assessed using flow cytometry and image processing in each well of Cytation ™ 3.

### 4.13. Cellular Imaging

Images of CD31^+^ cells were obtained using a Cytation 3 Cell Imaging multimode reader (BioTek Instruments, Inc., Winooski, VT, USA) tuned to DAPI, GFP and Texas Red light cubes.

At the end of the incubation period, lung endothelial cells were treated with fluorescent dyes Hoechst 33342, Annexin V-iFluor 350 and 7-AAD. Then, a Cytation 3 (magnification of 4× or 20×) was imaged, followed by cell analysis using Gen5 ™ data analysis software (Bad Friedrichshall, Germany).

All collected images were pre-processed to align the background before performing analytical methods. Cell analysis was performed on a blue channel to determine cell count based on the number of Hoechst stained nuclei. The default settings resulted in adequate calculated data for further analysis.

### 4.14. Statistical Analysis

Statistical analysis was performed using SPSS statistical software (version 15.0, SPSS Inc., Chicago, IL, USA). Data was analyzed and presented as means ± standard error of mean. A two-sided unpaired Student t-test (for parametric data) or Mann-Whitney test (for nonparametric data) was used according to distribution. A *p* value of less than 0.05 (by two-tailed testing) was considered an indicator of statistical significance.

## Figures and Tables

**Figure 1 ijms-20-01105-f001:**
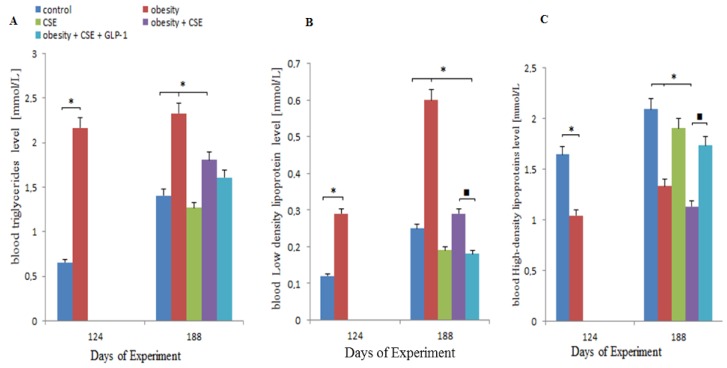
Lipid profile measurements in the blood of female C57BL/6 mice with obesity (the 124th day of the experiment) and obesity + CSE (the 188th day of the experiment): the level of triglycerides in serum (**A**); low-density lipoprotein level (**B**); high-density lipoprotein level (**C**). *—significance of difference compared with control (*p* < 0.05); ■—significance of difference compared with the obesity + CSE group (*p* < 0.05).

**Figure 2 ijms-20-01105-f002:**
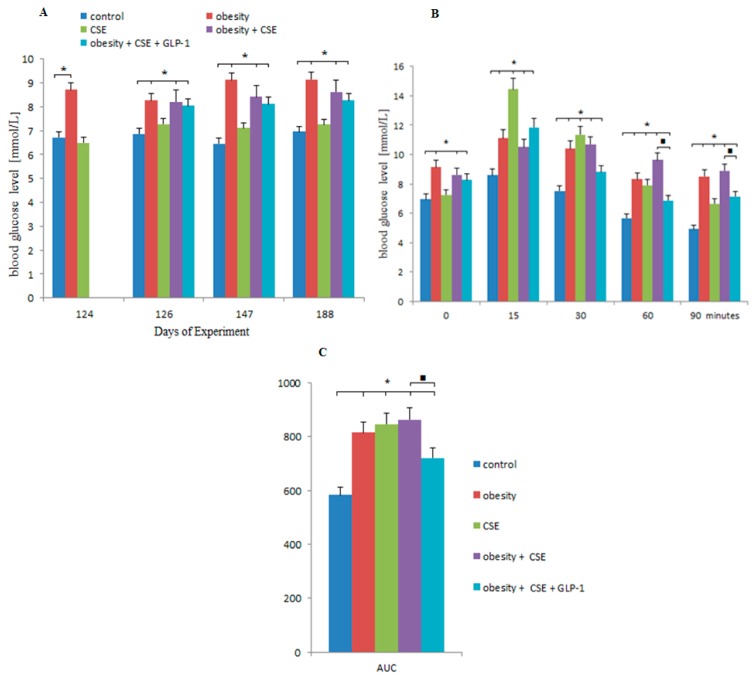
The glucose level in the blood of female C57BL/6 mice in the development of obesity and emphysema (**A**). Oral glucose tolerance test (**B**) and area under the curve (AUC) (**C**) on the 186th day of the experiment. *—significance of difference compared with control group (*p* < 0.05); ■—significance of difference compared with the obesity + CSE group (*p* < 0.05).

**Figure 3 ijms-20-01105-f003:**
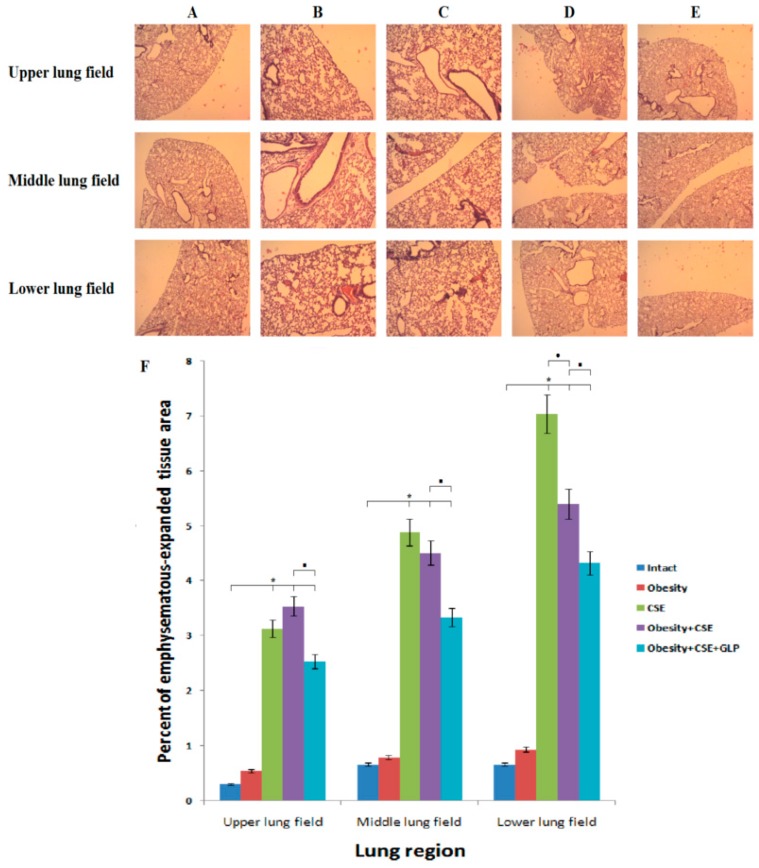
Photomicrographs of left lung sections (upper, middle and lower pulmonary field) obtained from female C57BL/6 mice (*n* = 6). Tissues were stained with hematoxylin-eosin (188th day of the experiment). (**A**) Control group; (**B**) Obesity; (**C**) CSE; (**D**) Obesity + CSE; (**E**) Obesity + CSE + GLP-1; (**F**) The area of emphysema-expanded lung tissue of mice from all groups. * *p* < 0.05 significance of difference compared with control group, ■ *p* < 0.05 significance of difference compared with the obesity + CSE group.

**Figure 4 ijms-20-01105-f004:**
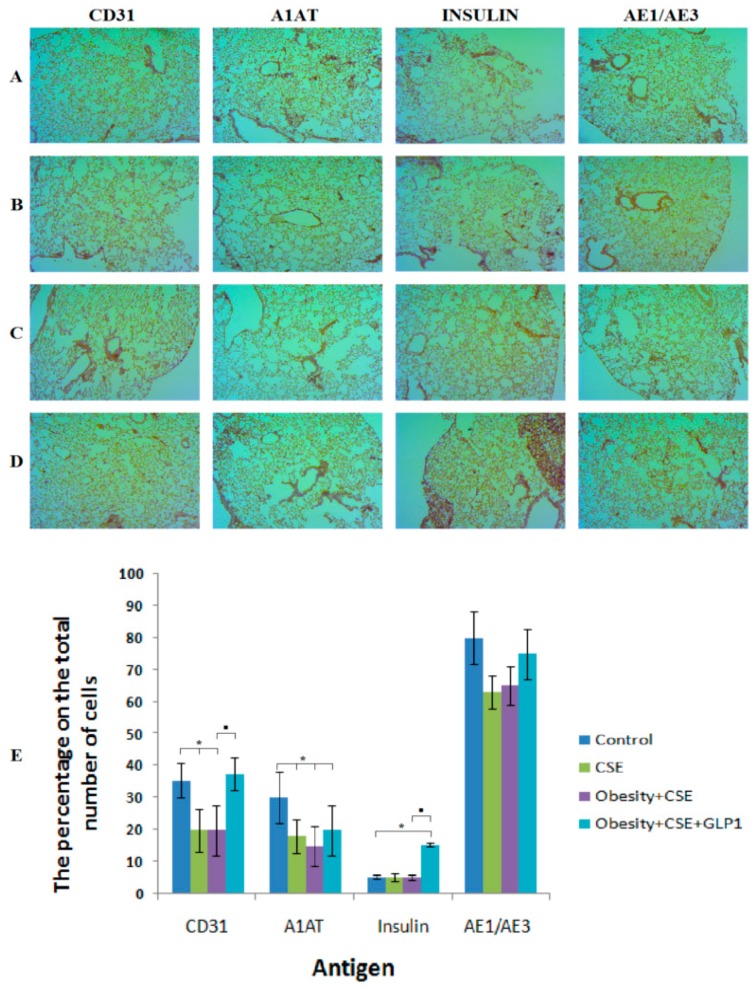
Immunohistochemical staining for specific cellular markers: CD31, α1-antitrypsin (A1AT), insulin, pan-cytokeratin (AE1/AE3) in the lungs from female C57BL/6 mice (brown stain). (**A**) Control group; (**B**) CSE; (**C**) Obesity + CSE; (**D**) Obesity + CSE + GLP-1. (**E**) The relative content of cells expressing specific antigens in all groups. * *p* < 0.05 significance of difference compared with control group, ■ *p* < 0.05 significance of difference compared with the obesity + CSE group.

**Figure 5 ijms-20-01105-f005:**
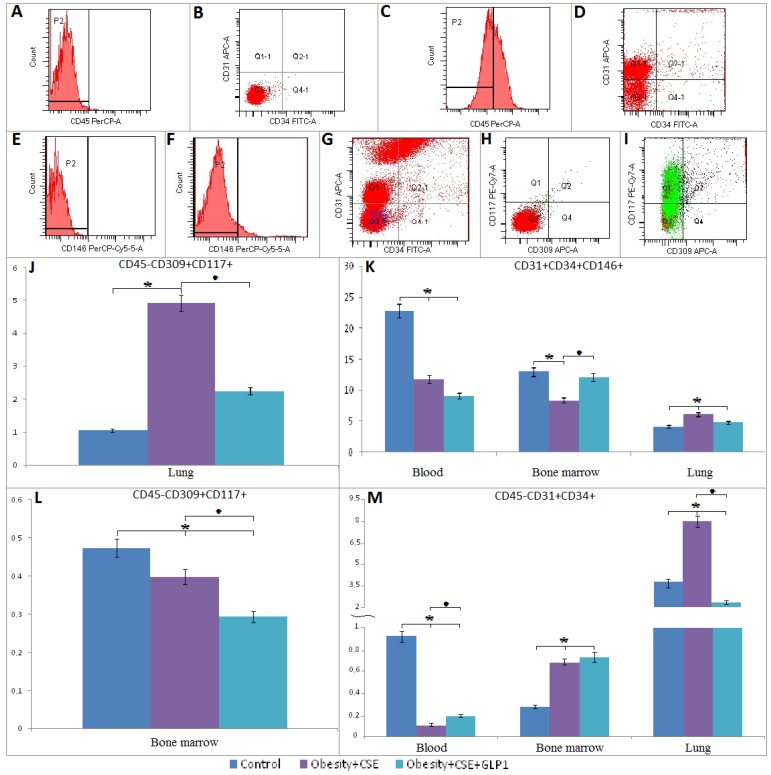
Characterization of endothelial cell population CD31^+^CD34^+^CD146^+^, CD45^−^CD31^+^CD34^+^, CD45^−^CD117^+^CD309^+^ isolated from blood, bone marrow, and lung of female C57BL/6 mice on the 188th day of the experiment. Cells were analyzed by flow cytometry using antibodies for CD45, CD31, CD34, CD146, CD117, CD309 mice. Dot plots are representative for three independent experiments with the mean from three independent experiments. (**A**,**B**) Isotypic control for IgG2a (PerCP), IgG2a (FITC) and IgG2b (APC). (**C**,**D**) Phenotype establishment and qualitative analysis of CD45 (PerCP), CD34 (FITC) and CD31 (APC) expression. (**E**) Isotypic control for IgG2a (PerCP-Cy5.5). (**F**,**G**) Phenotype establishment and qualitative analysis of CD34 (FITC), CD31 (APC), and CD146 (PerCP-Cy5.5) expression. (**H**) Isotypic control for IgG2b (PE-Cy7), and IgG2b (APC). (**I**) Phenotype setting and qualitative analysis of CD45 (PerCP), CD117 (PE-Cy7), and CD309 (APC) expression. (**J**,**L**) The content of angiogenesis progenitor cell (CD45^−^CD117^+^CD309^+^) in the lung (**J**) and bone marrow (**L**) of control mice, obese mice exposed to CSE, obese mice exposed to CSE and receiving GLP-1. (**K**) The content of endothelial cells with CD31^+^CD34^+^CD146^+^ phenotype isolated from blood, bone marrow, and lung of control mice, obese mice exposed to CSE, obese mice exposed to CSE and receiving GLP-1. (**M**) The content of endothelial cells with phenotype CD45^−^CD31^+^CD34^+^ isolated from blood, bone marrow, and lung of control mice, obese mice exposed to CSE, obese mice exposed to CSE and receiving GLP-1. *—significance of difference compared with control (*p* < 0.05); ■—significance of difference compared with the obesity + CSE group (*p* < 0.05).

**Figure 6 ijms-20-01105-f006:**
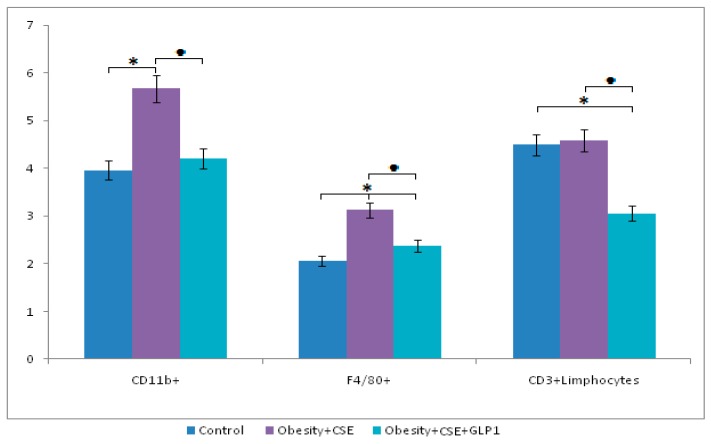
The content of CD11b^+^ and F4/80^+^ monocytes, and CD3^+^ T-lymphocyte in the blood of control mice, obese mice exposed to CSE, obese mice exposed to CSE and receiving GLP-1. *—significance of difference compared with control (*p* < 0.05); ■—significance of difference compared with the obesity + CSE group (*p* < 0.05).

**Figure 7 ijms-20-01105-f007:**
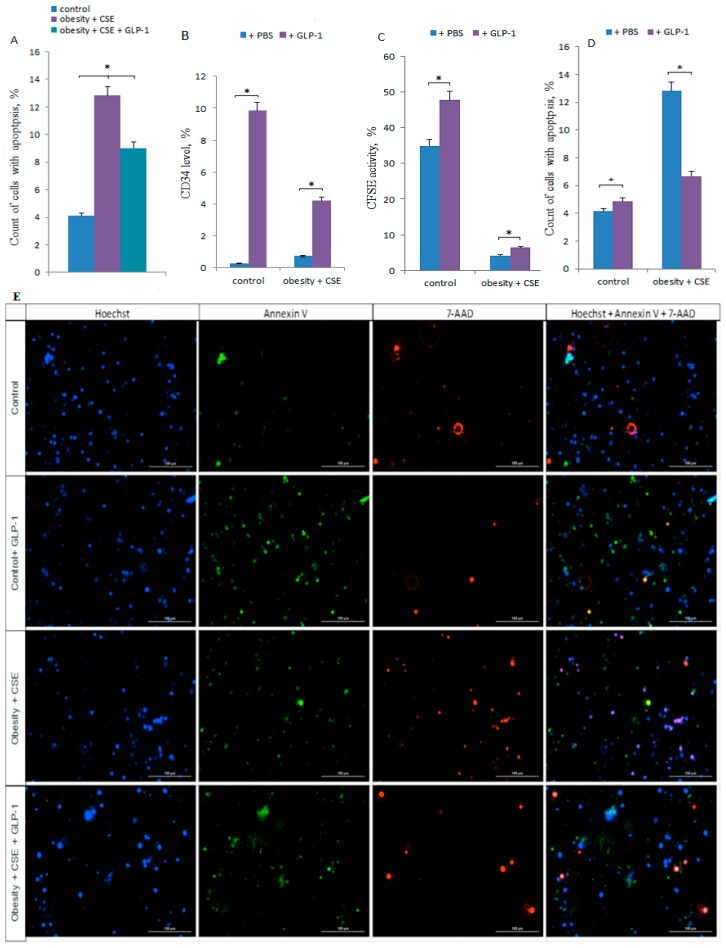
GLP-1 treatment effects on CD31^+^ endothelial cells isolated from the lungs of female C57BL/6 mice. (**A**) The count of CD31^+^ cells with apoptosis in the lungs of female C57BL/6 mice on the 188th day of the experiment. *—significance of difference compared with control (*p* < 0.05); ■—significance of difference compared with the obesity + CSE group (*p* < 0.05). (**B**–**D**) CD31^+^ endothelial cells from lung were precultured for 5 days, incubated with or without GLP-1 (10-7 M) for 24 h and then labeled with Hoechst, CD34 FITC (**B**), CFSE (**C**), Annexin V and 7-AAD (**D**) prior to fluorescence microscopic analysis. (**B**) The level of CD34^+^ cells after culture, (**C**) CFSE activity after culture, (**D**) the count of cells with apoptosis after culture. All data are expressed as mean ± SD, *—significance of difference compared with control without GLP-1 (*p* < 0.05). (**E**) 20× images of CD31^+^ cells stained with: Hoechst (blue) to identify cell nuclei; Annexin V (green); 7-AAD (red); (Hoechst + Annexin V + 7-AAD) composite image using all three colors. Determination of the percent of cells in apoptosis is made by the ratio of cells counted in green and red channel to total cells counted using blue (DAPI) channel. All scale bars are 100 µm.

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
