# Peer review of "Endothelial Progenitor Cells as Pathogenetic and Diagnostic Factors, and Potential Targets for GLP-1 in Combination with Metabolic Syndrome and Chronic Obstructive Pulmonary Disease"

_ijms, 2019, doi:10.3390/ijms20051105_

Round 1

Reviewer 1 Report

In this manuscript titled, " Endothelial Progenitor Cells as Pathogenetic and Diagnostic Factors, and Potential Targets for GLP-1 in Combination with Metabolic Syndrome and Chronic Obstructive Pulmonary Disease ", Evgenii Germanovich Skurikhin et al., authors examine the effectiveness of the GLP-1 treatment in inflammation and emphysema, incretin stimulation of endothelium and epithelium regeneration of the alveolar  tissue in the C57BL/6 mice with lipid disorders and emphysema. Authors presented evidence of the effectiveness of treatment with incretin GLP-1 for the symptoms of obesity and diabetes, inflammation and emphysema, as well as stimulation of regeneration of damaged endothelium of the lungs in conditions of modeling obesity and emphysema. They proposed CD31 and CD34 positive endothelial progenitor cells were a potential target of GLP-1. This manuscript is well written. For the study, the presented data are quite sufficient.

Author Response

Dear Reviewer#1

Thank you for your review. We are very grateful to the reviewer for your work in reviewing our manuscript. We revised the manuscript. Below please find the reviewers’ comments and our responses. The elucidations and the changes have been included into the revised version.

Author's Responses to Questions

Reviewer #1:

Comments and Suggestions for Authors

“In this manuscript titled, " Endothelial Progenitor Cells as Pathogenetic and Diagnostic Factors, and Potential Targets for GLP-1 in Combination with Metabolic Syndrome and Chronic Obstructive Pulmonary Disease ", Evgenii Germanovich Skurikhin et al., authors examine the effectiveness of the GLP-1 treatment in inflammation and emphysema, incretin stimulation of endothelium and epithelium regeneration of the alveolar tissue in the C57BL/6 mice with lipid disorders and emphysema. Authors presented evidence of the effectiveness of treatment with incretin GLP-1 for the symptoms of obesity and diabetes, inflammation and emphysema, as well as stimulation of regeneration of damaged endothelium of the lungs in conditions of modeling obesity and emphysema. They proposed CD31 and CD34 positive endothelial progenitor cells were a potential target of GLP-1. This manuscript is well written. For the study, the presented data are quite sufficient”.

Yours sincerely,

Prof. Evgenii Skurikhin

Head, Laboratory of Regenerative Pharmacology 

Goldberg Research Institute of Pharmacology and Regenerative Medicine, Tomsk National Research Medical Centre of the Russian Academy of Sciences (dir. – prof. V.V. Zhdanov)

Lenin Street, 3, Tomsk, Russia

634028

Fax: +7-(3822)-418-375

E-mail: eskurihin@inbox.ru

Reviewer 2 Report

Comments to Authors   

The current study: 1) have presented evidence of the effectiveness of treatment with incretin GLP-1 for the symptoms of obesity and diabetes, inflammation and emphysema, as well as stimulation of regeneration of damaged endothelium of the lungs in conditions of modeling obesity and emphysema; 2) have proposed a potential target of GLP-1: expressing markers of CD31 and CD34 endothelial progenitor cells.

                Several epidemiologic studies have implicated visceral fat as a major risk factor for insulin resistance, type 2 diabetes mellitus, cardiovascular disease, stroke, metabolic syndrome, COPD and death [1]. Glucagon-like peptide-1 (GLP1) is an important insulin secretagogue that possesses antiinflammatory effects. GLP1 receptor (GLP1R) agonists have been demonstrated to serve a pivotal role in the treatment of obstructive lung diseases, including chronic obstructive pulmonary disease (COPD) [2].  It was demonstrated that GLP‑1R overexpression markedly suppressed IL‑1β, IL‑4, TNF‑α and GM‑CSF levels. GLP‑1R overexpression upregulated the expression levels of adenosine triphosphate‑binding cassette, subfamily A, member 1 (ABCA1) in ASM cells, and the effects of GLP‑1R on cell proliferation and migration, and inflammatory cytokine expression in ASM cells was abolished by siRNA‑mediated silencing of ABCA1 [2]. Another, it was suggested that GLP‑1R contributes to COPD pathology, potentially via an ABCA1‑mediated pathway [2]. Recently, it was showed that GLP-1R activation protects mice from LPS-induced acute lung injury (ALI) by maintaining functional endothelial barrier and inhibiting PMN extravasation [3]. Another, it was suggested that GLP-1R may be a potential therapeutic target for the treatment of ALI [3].

                  Authors are kindly requested to emphasize the current concepts about these issues in the context of recent knowledge and the available literature. This article should be quoted in the References list.

References

Should visceral fat be      reduced to increase longevity? Ageing Res Rev. 2013 Sep; 12 (4): 996-1004.      doi:10.1016/j.arr.2013.05.007.

Overexpression of GLP-1      receptors suppresses proliferation and cytokine release by airway smooth muscle      cells of patients with chronic obstructive pulmonary disease via activation      of ABCA1. Mol Med Rep. 2017 Jul; 16 (1): 929-936. doi:10.3892/mmr.2017.6618.

Glucagon-like peptide-1      receptor activation alleviates lipopolysaccharide-induced acute lung injury      in mice via maintenance of endothelial barrier function. Lab Invest. 2019 Jan      18. doi: 10.1038/s41374-018-0170-0.

Author Response

To the Reviewer #2

Dear Reviewer#2

Thank you for your review. We are very grateful to the reviewer for your work in reviewing our manuscript. We revised the manuscript. Below please find the reviewers’ comments and our responses. The elucidations and the changes have been included in the revised version.

Author's Responses to Questions

Reviewer #2:

Comments to Authors   

The current study: 1) have presented evidence of the effectiveness of treatment with incretin GLP-1 for the symptoms of obesity and diabetes, inflammation and emphysema, as well as stimulation of regeneration of damaged endothelium of the lungs in conditions of modeling obesity and emphysema; 2) have proposed a potential target of GLP-1: expressing markers of CD31 and CD34 endothelial progenitor cells.

Several epidemiologic studies have implicated visceral fat as a major risk factor for insulin resistance, type 2 diabetes mellitus, cardiovascular disease, stroke, metabolic syndrome, COPD and death [1]. Glucagon-like peptide-1 (GLP 1) is an important insulin secretagogue that possesses anti-inflammatory effects. GLP 1 receptor (GLP 1R) agonists have been demonstrated to serve a pivotal role in the treatment of obstructive lung diseases, including chronic obstructive pulmonary disease (COPD) [2].  It was demonstrated that GLP 1R overexpression markedly suppressed IL 1β, IL 4, TNF α and GM CSF levels. GLP 1R overexpression upregulated the expression levels of adenosine triphosphate-binding cassette, subfamily A, member 1 (ABCA1) in ASM cells, and the effects of GLP 1R on cell proliferation and migration, and inflammatory cytokine expression in ASM cells was abolished by siRNA mediated silencing of ABCA1 [2]. Another, it was suggested that GLP 1R contributes to COPD pathology, potentially via an ABCA1 mediated pathway [2]. Recently, it was showed that GLP-1R activation protects mice from LPS-induced acute lung injury (ALI) by maintaining a functional endothelial barrier and inhibiting PMN extravasation [3]. Another, it was suggested that GLP-1R may be a potential therapeutic target for the treatment of ALI [3].

Authors are kindly requested to emphasize the current concepts about these issues in the context of recent knowledge and the available literature. This article should be quoted in the References list.

References

1. Should visceral fat be reduced to increase longevity? Ageing Res Rev. 2013 Sep; 12 (4): 996-1004.      doi:10.1016/j.arr.2013.05.007.

2. Overexpression of GLP-1      receptors suppresses proliferation and cytokine release by airway smooth muscle cells of patients with chronic obstructive pulmonary disease via activation of ABCA1. Mol Med Rep. 2017 Jul; 16 (1): 929-936. doi:10.3892/mmr.2017.6618.

We have added articles in the References list.

Yours sincerely,

Prof. Evgenii Skurikhin

Head, Laboratory of Regenerative Pharmacology 

Goldberg Research Institute of Pharmacology and Regenerative Medicine, Tomsk National Research Medical Centre of the Russian Academy of Sciences (dir. – prof. V.V. Zhdanov)

Lenin Street, 3, Tomsk, Russia

634028

Fax: +7-(3822)-418-375

E-mail: eskurihin@inbox.ru